



# Misinterpretation of hydrological studies in the Lancang-Mekong Basin: drivers, solutions and implications for research dialogue

Wenling Wang[1], Richard Grünwald[1], Yan Feng[2]

[1]Institute of International River and Eco-Security/Asian International Rivers Center, Kunming, 650091, China
[2]Yunnan Key Lab of International Rivers and Transboundary Eco-security, 650091, Kunming, China

*Correspondence to*: Wenling Wang (Wang.wangwl@ynu.edu.cn), Richard Grünwald (grunwaldrichard@ynu.edu.cn), Yan Feng (Fengyan@ynu.edu.cn)

**Abstract.**

Socio-hydrology presents one of the scientific approaches interpreting complex interactions between human and water systems. To date, water becomes extremely politicized by non-scientists and frequently put in a broader political context with non-water issues. The purpose of this text is to (1) analyse drivers of the growing politicization of hydrological science in the Lancang-Mekong Basin, (2) examine solutions for addressing the misinterpretation of hydrological data, and (3) outline the unintended consequences of politicization the hydrological studies. The paper argues that politicization of science (i) gives

more power to non-scientists, (ii) undermines the trust in science and other research institutions, (iii) creates inequality among hydrological studies and water scientists, and (iv) provides more incentives for making research tailored to desirable outcomes. The topic is highly actual and beneficial for water experts and other scientists who want to better understand the potential negative implications of hydrological studies and the limits of socio-hydrology.

## Introduction

Water can be interpreted in many ways. While the majority of hydrological interpretations consider water as an apolitical exploitable renewable source satisfying the growing demand for human development, water issues are becoming increasingly politicized by various non-scientific approaches in recent years. Perhaps, the most interesting trend how hydrological studies may deepen the conflict of interests among multi-stakeholders present the politicization of science in the Lancang-Mekong Basin. To date, there are many myths simplifying the research conclusions and even more information gaps in understanding

the complex interactions between human and water systems. However, what feeds the uncertainty the most is the interpretation beyond data, different quality and purpose of the hydrological studies that may create a series of controversies and polarize both scientific and non-scientific audience. In this paper, we use the socio-hydrology perspective to understand the "power" of research studies regardless of their theoretical-methodological shortcomings, illustrate broader political implications for misinterpreting the hydrological studies to justify certain water narratives and describe current dilemmas in conflict resolution.



The paper is divided into a few sections. In section 1, we briefly outline socio-hydrology and describe its connections with the politicization of science. In section 2, we explore the drivers and techniques leading to the politicization of science and study the accountable solutions to address the conflict of interests among multi-stakeholders within the Lancang-Mekong Basin. In section 3, we focus on two hydrological studies (Pöyry Report and Eyes on Earth Study) that were used as a political tool to justify certain national interests and analyze their implications on the existing water cooperation. The main attention will be

paid to their impact on the transboundary water governance and existing water cooperation mechanisms responsible for hydrological data sharing and other water-related research, particularly the Lancang-Mekong Cooperation (LMC) and the Mekong River Commission (MRC). The last part of the text summarizes the conclusions from the research and propose recommendations for minimizing the misunderstandings and widespread misinterpretation of the hydrological data.

## 1. Socio-hydrology and politicization of science

Socio-hydrology has been firstly introduced in 2012 as a "new science" linking hydrology with socio-economic issues (Sivalapan et al. 2012). Similarly like other interdisciplinary approaches, socio-hydrology analyses various socio-hydrologic phenomena and explores the complex human-water interactions. While early socio-hydrological works mainly focused on understanding the dynamics and visualization of the co-evolution of the coupled human-water systems (Baldassare et al. 2013a, Baldassare et al. 2013b, Elshafei, et al. 2014), some authors believe that socio-hydrology only draws on the hydro-sociology

concept proposed by Falkenmark (Falkenmark 1979), and use sociology as an intuitive tool to verify causal relations, describe illogical behaviour of multi-stakeholders and/or acknowledge other factors (e.g. Madani and Shafiee-Jood 2020: 6, Evers et al. 2017). Because most of the socio-hydrological studies use quantitate methods to grasp complex connections and evaluate comprehensive trajectories based on predictive mathematical models (Sivalapan et al. 2012, Seidl and Barthel 2017), many scholars test the qualitative non-scenario based approaches and include less predictable variables, including uncertainty,

politics and power. Another interesting pathway for the socio-hydrology present interpreting the existing water-related narratives over transboundary water resources (Ert, Cohen-Amin and Dinar 2019). To date, there can be identified many water paradigms which are shaped by cognitive bias and other subjective assumptions. While some narratives represent self-fulfilling theories (e.g looming water crisis and water wars as a result of water scarcity) (e.g. Gleick 1998), socially constructed scenarios (e.g. for every water cooperation there must be an agreement prescribing the transboundary water management rules) (e.g.

Yoffe, Wolf, and Giordano 2003), other ones are revising binary way of thinking (e.g. false assumption that all conflicts are "bad" and all cooperations are "good" and there are only positive and negative implications from hydro-hegemony) (Cascão a Zeitoun 2010, Zeitoun and Mirumachi 2008) and reconsidering existing behaviour patterns (e.g. recognizing various degrees of cooperation and conflict and accepting co-existence of cooperation and conflict) (Zeitoun et al. 2017; Earle, Jägerskog & Öjendal, 2010; Wegerich & Warner, 2010; Cascão & Zeitoun, 2010).

Unlike these approaches interpreting the water issues, socio-hydrology was mainly developed among natural scientists, particularly hydrologists (Octavianti and Charles 2019) which predominantly reflect positivist, technocratic and utilitarian



paradigms, the socio-hydrology continues to expand to new directions. Given its presents trajectory, many scientists blame political officials, media and other external actors for misinterpretation of their work (i.e. scientists vs. non-scientists). However, the alienation of science is also driven by researchers self-interests, sympathies and other socially constructed

arguments (i.e. scientists vs. scientists) supporting certain socio-hydrological narratives. In the era of disinformation, the co-evolution between scientists and non-scientists is highly competitive and frustrating for both sides. Also, while the lay knowledge is predominantly considered less valid and one of the factors for unsustainable development (e.g. Falkenmark et al. 1999), the involvement of the civil society and grass-root organizations may be useful to replicate local scientific knowledge and design better adaptation strategies for water resources management (Sneddon, Magilligan and Fox 2017). Perhaps, the

most successful examples of using "citizen science" can be traced to Thai Baan research (Mekong Watch 2005) or the River Chiefs system (Li, Tong and Wang, 2020, Liu et al. 2019) that provides immense potential for further involvement of non-state actors in the decision-making process. On the other hand, in contrast with traditional scientific thinking, the lay people tend to deconstruct the supremacy of scientists who often ignore their knowledge by engaging in various public activities (Weng 2015). While their passion to see the immediate changes and practical results of their work may be widely considered

as a reckless attempt to finding the "shortcuts in hydrological science", such volunteers with lower technical and intellectual capacities may still collect valuable data, monitor the local environments, raise public awareness about the current water issue, support various restoration projects, build trust with local communities and otherly help the water professionals with their ongoing research. Also, the volunteers are usually budget-friendly, time flexible, more enthusiastic for innovations and eager to help especially if they serve the greater good of a society that their research counterparts. However, without the legal code

of conduct, the non-state actors will only enjoy enlarging the number of privileges without taking more responsibilities for their actions in transboundary water governance (Yasuda 2018).

Another interesting groups where non-scientists play an important part in water management are politicians and the justification of various infrastructure water projects. The water projects, particularly hydropower dams are historically considered as a symbol of national pride, prosperity and the technical progress in conquering Nature (Molle, Mollinga and

Wester 2009: 332-334; Menga and Mirumachi 2016: 378; Menga 2015). While human mankind always try to transform the local environments to serve a better purpose for human societies, grandiose water projects always attract significant media attention. Until the late 1990s, the multipurpose hydropower projects were widely considered as a (i) cheap, renewable and "clean" source of energy, (ii) effective tool for regulating the water flow, and (iii) a socio-economic radix promoting people's livelihood and human society development (Mayeda and Boyd 2020; Grünwald 2018). With growing opposition against the

rapid hydropower development, the civil society in various corners of the world often calls for small hydropower plants and other renewable sources of energy, particularly solar and wind power, that are often perceived environmentally friendly and less likely to be a subject of public protests (Venus et al. 2020). However, despite fabulating the small-scale hydropower projects and criticizing various financial institutions (e.g. World Bank and Asian Development Bank) for funding the controversial dams (Matthews and Geheb 2015: 8), the small and medium-size hydropower dams attract diametrically fewer

investments and political prestige than large-scale hydropower dams (Yang 2001; Daojing 2015: 40).



Last but not least specific group of stakeholders represent the subversive actors (Grünwald and Kouřil 2018). These actors represent local mafias, smugglers, guerrilla groups but also other representatives who can oppose further control over the watersheds (e.g. lack of central government's control over the China-Myanmar borderlands in the Golden Triangle), promote specific infrastructure along the riverbeds (e.g. creating special economic zones within the international basins may also

casinos, hotels and other facilities connected to money laundering activities) or spreading fears over the infrastructure safety (e.g. debates about the weaponization and other water terrorist attacks on water infrastructure and systems) (Grünwald 2018, Gleick 2004, Gleick 2006). However, such adverse impacts on sustainability water resources may also arise by intentional water mismanagement (Butler, Scammell and Benson 2016; Boelens, Getches, Guevaragil 2010), restricting the access to water for local communities (Milton et al. 2017, Mehta 2011) or unilateral utilization of transboundary waters (Vasilenko

2017, Zhong et al. 2016, Gleick 1993). Therefore, whenever the socio-hydrologists emphasizing the positive outcomes from better utilization of shared waters that may eventually outweigh the negative consequences of water over-exploitation, water issues is never fully apolitical. In fact, by legitimizing the rapid development of international rivers and marginalizing the associate cost of the river development (Aradau 2004: 402; Buzan and Hansen 2009: 216-217), state water authorities may ignore existing social inequalities among sectors and justify negative environmental impacts as an unavoidable cost for

ensuring the national water security in the name of technological and economical solutions (Matthews and Motta 2015: 6271). On the other hand, the inevitable conflict of interests among multi-stakeholders or disruption of existing water systems does not necessary to be negative as some researchers suggest (Ravnborg 2004). Instead, the multi-level tension can expose weak spots in cross-sector water collaboration, identify shortcomings in existing legislative or find various ineffective water utilization habits that can be bureaucratically and scientifically changed.

**2. Drivers, techniques and degrees misinterpretation of hydrological science**

According to a survey related to the politicization of science published in 2016, at least half of the scientific papers can be trusted (Baker 2016). Drawing on the survey results, authors concluded that the crisis of science is mainly caused by (i) selective reporting, (ii) pressure to publish, (iii) low statistical power or poor analysis, (iv) lack of replicability, (v) insufficient oversight/mentoring, (vi) unclear methodology, (vii) poor experimental design, (viii) lack of primary data, (ix) fraud, and (x)

insufficient peer review. Such conclusions are not surprising by considering the syndrome of duelling experts (Wade 2004) where various multi-stakeholders contests their ideas and solutions. While the diversity of professional opinions on better water resources management may positively enhance the cross-sectorial collaboration, reaching the consensus among various water experts can be sometimes very difficult and time-consuming regarding different scientific background (Earle, Jägerskog and Öjendal 2010, Swann and Bosson 2008). However, because scientists are mainly focusing on understanding various

degrees of uncertainty (Dietz 2013) and seeking to innovate solutions regardless of the prevalent research paradigms (Brown 2015: 21), full trust in science can be troublesome. Similarly, like any politicized scientific narratives affecting the credibility



of scientists across different fields (e.g. climate change deniers, genetical modification alarmists or anti-vaccination movements), there are several drivers which feed the contemporary water narratives.

Firstly, there is an issue of individual experience and the illusion of superiority of personal opinion. If the consequences are more immediate, vivid and repeatedly experienced, the incentives for the politicization of science and simplifying the complex issues, particularly in public media seems to be higher (Kreps and Kriner 2020, Petersen, Vincent and Westerling 2019). The biased judgement may depend on (i) intellectual confidence (i.e. Dunning-Kruger effect highlighting the power of common sense and limited knowledge vs. Imposter Syndrome emphasizing the uncertainty and over-information), (ii) level of education (i.e. the degree of wisdom and critical thinking for a deep understanding of the issue), (ii) cultural, religious and other personal values (i.e. sympathies with certain stakeholder views, obsession with proofing certain hypothesis), (iii) political cues (i.e. frequency of medialization of indirect non-water remarks in a broader political context to shape the current water narratives) and (iv) various external factors, including the media hyping and other disinformation efforts (e.g. Lyengar and Massey 2019; Petersen, Vincent and Westerling 2019). However, the biggest challenge still represents the anti-scientific and non-scientific approaches that exacerbate science communication and develop their arguments beyond science (Morgan et al. 2018). Unlike in constructive dialogues, these approaches use a variety of tools to discredit other's arguments. From whataboutism (i.e. evading and re-directing the discussion to other topics), hypocrisy (i.e. showing higher moral superiority to justify double-standard in discussion), selective objections (i.e. devaluating whole research study by pointing out the theoretical-methodological inconsistencies), false accusations (i.e. once the slander is used to damage someone's reputation, it requires more effort to refute the opposite claims regardless the scientific achievements and facts) to populism (i.e. proposing emotionally pleasing solutions and strict labelled actions that will change the existing water-challenges with minimal effort and immediate results overshadow the traditional mechanisms for conflict resolutions) and other argumentation manoeuvrings (Aspeitia 2020, Fritz, Kyle and Miller 2018, van Laar 2007).

Secondly, while some water narratives may intentionally alienate existing research knowledge and draw conclusions beyond data, others may appear as a result of unintended consequences (Merton 1936). At this point, the main reason for misinterpretation of hydrological data is driven by (a) ignorance (e.g. estimating the trends with lack of hydrological data), (b) human errors (e.g. selection of key parameters in research analysis), (c) personal interests (e.g. promoting "water security for whom"), (d) change of values (e.g. promoting commercial cash-crops over traditional agriculture) and (e) self-fulfilling prophecies (e.g. concerns about using trickles down diplomacy by upstream countries) (Thu & Wehn, 2016, Merton, 1936; Boudon 1977). Such unintended consequences can be also traced (i) exponential growth of science (i.e. widening the socio-hydrological agenda and over-information creates more uncertainties), (ii) diversification of funding sources (i.e. governmental funds and non-governmental research projects may support both highly innovative and biased research), (iii) research labelling (i.e. western scientists and "independent" research institutions do not automatically guarantee "more credible" research), (iv) research transparency (i.e. public availability of sensitive datasets and conclusions may both encourage the local knowledge development but also polarize the society) and/or (v) concerns from public and political response (i.e. publishing groundbreaking discoveries may diversify the research understanding and provide guidelines for future decision-making but



also lead scientists to undermine their science by self-censorship to adapt "more desirable" opinions according to actual academic, public and political environment) (Morgan et al. 2018, Spruijt et al. 2014, Lupia 2013, Pielke 2004).

Another factor affecting the escalation and de-escalation of conflict of interests over science lies in the social tendency of putting water issues into the realm of politics. Firstly, there is the politicization of apolitical scientific knowledge that was previously out of the public domain (Buzan, Weaver and Wilde 1998). At this stage, the water issues (e.g. severe droughts, dam collapse) that were predominantly interpreted as a natural cause without any political connotations (Zeitoun and Mirumachi 2008) and vaguely acknowledged both by state officials and the wide public (Grünwald, Feng and Wang 2020: 9) become part of broader political agenda (Oosterloo 2016) by increasing public awareness, feeding the prejudice against political opponents and blurring the boundaries between scientific and non-scientific assumptions to justify certain stakeholder's interests (Matthews and Geheb 2015, Zeitoun et al. 2017). Secondly, there is the re-politicization of already politicized water-issues that are periodically under scrutiny from both scientists and non-scientists (Buzan, Weaver and Wilde 1998). In this case, the discussion over the water issues invokes series of civil protests as a result of incapability or unwillingness to solve the problem by state authorities (Grünwald, Wang, Feng 2020: 5). However, any attempt to change the existing political discourse and move back the water issues out of the national agenda (e.g. produce more joint research studies, giving more promises and political guarantees) (Jessop 2014, Atkins 2019) often backfire and lead to even more systemic marginalization of water issues (e.g. ignoring the independent research reports and reflecting only the lawsuits and privileged actors). Thirdly, there is the de-politicization of water issues where both scientists and non-scientists put their faith in "purely technical solutions" (Cuttita 2018, Jessop 2014). Despite the criticism for delaying the conflict resolutions and doing "too little, too late", there is still a constructive dialogue on how to solve the actual water issues among multi-stakeholders and strong confidence in political authorities that can keep promises and provide feasible compromises (Grünwald, Wang and Feng 2020: 5).

## 2.1. Solutions for de-politicization of science

Currently, there are several ways how water narratives can be demythized. Firstly, there are official channels through which governmental institutions and other authorized multi-stakeholders, including designated water institutions, universities, financial institutions and other assistant companies held the structured dialogue about the current trends, existing challenges and potential conflict resolutions. Such accountable research and policy discussions (Kerckhove, Rennie and Cormier 2015) are either held privately (e.g. closed-door negotiations with small groups of designated water professionals) or publicly (e.g. regional stakeholders meetings, scientific conferences, workshops and other research events). These interactions are usually well-organized and planned to deal with the complex water agenda and manage the invitation for numerous researchers from different fields. However, the official channels are usually (i) time-consuming, (ii) narrowly specialized (i.e. the dominance of several socio-hydrological perspectives), and (iii) outcome-oriented (i.e. focus on reaching consensus rather than building mutual trust and better understanding). Additionally, if the data are too sensitive or confidential (Gleick 1993: 98) to discuss





them in public, it is even more difficult to draw firm conclusions regarding the limited research audience. On the other hand, by recognizing the source of the problem and finding the common ground, there is a bigger chance to address the existing

misinterpretation and implement the revisions about current water paradigms through various official speech acts, national campaigns and other strategies. However, if the state authorities decide to use legislative tools and other measures to delete or marginalize certain information, the outcome for combating against misinterpretation can be the opposite (Jansen and Martin 2015).

Secondly, there are semi-formal communication channels where various researchers (e.g. research institutions, universities,

think-tanks) consult and discuss their approaches and discoveries with other experts. Such "scientists-for-scientists" dialogues can be specially identified in terms of (a) annual research conferences, (b) specialized workshops and (c) other academic forums (Rowlands et al. 2011). The special category represents the research journals, policy studies, and other indexed papers ensuring a certain degree of constructive dialogue and basic research standards (i.e. peer-review process in high-rating journals, transparent methodology and datasets). While not all researchers possess enough experience nor credibility to be invited into

official channels, this level of communication is crucial to confront, enrich and verify own research conclusions with a broader academic audience (Ware 2008). However, there is a great asymmetry between hydrological studies, particularly in terms of (i) quality of the work (e.g. missing datasets, impossible replicability of research results, lack of peer-review process), (ii) theoretical-methodological perspective (e.g. using different methods how to calculate the same phenomenon with different accuracy, theoretical flaws in operationalized concept), (iii) affiliation (e.g. faith in western scientists or prestige research

institutions), (iv) preferable language (e.g. dominance of English written articles) and (v) audience (e.g. difficulty to understand research jargon by non-scientists) (Colquhoun 2011, Bohannon, 2013). However, there are still many predator and other research journals that do not fact-check the content and accept the research papers with duplicate content, controversial data and other texts with empirical shortcomings in exchange for mandatory fees for publishing (Vaikl 2019).

Another specific sub-category represents various assistant companies, information agencies and other fact-checking

individuals collaborating with the government to fight against the disinformation. While their activities may range from monitoring the existing myths, mapping the sources of misinterpretation, designing disinformation campaigns, communicating with public society and providing the recommendations "not-to-do list" for policy-makers and other state institutions (Haciyakupoglu et al. 2018, Lim 2018). However, the effectiveness of fact-checking organizations significantly varies (Pavleska et al. 2018).

Thirdly, there are informal communication platforms like (a) civil-society dialogues (e.g. NGOs and other non-state actors events), (b) social networks (e.g. Twitter, Facebook, WeChat), (c) public and private media (e.g. blogs, opinions) and other unofficial events (e.g. roundtables, webinars) that may put pressure on local authorities and researchers (Rowlands et al. 2011). Although informal channels present flexible, accessible and popular non-scientists friendly platforms for interpreting the existing water narratives, it is more challenging to (i) ensure the credibility of facts, (ii) understand the existing limits and key

message of referred research, and (iii) ensure constructive feedback and structured dialogue between multi-stakeholders. Also, despite many scientists and state officials use these platforms to restore faith in science and clarifying existing myths in the





water sector, informal networks may also lead to "information trench wars" and further politicization of science supporting certain narratives which is why researchers are still quite conservative to this type of communication (Bergner 2010). On the other hand, scientific blogging and research networking activities may positively deepen the discussion and provide valuable

feedback that may eventually lead to future publications. However, thanks to lack of incentives from research institutions, linguistic barriers, concerns about negative reviews, media marketing context and less strict rules for high-quality texts, social media still remains complementary rather than a substitute to traditional research channels (Martin and MacDonald 2020).

## 2. Politicization of science in the Lancang-Mekong Basin

Within the Lancang-Mekong Basin, there can be identified two main drivers for the politicization of hydrological science.

Firstly, there is wide ignorance of hydrological science and other impact assessment studies to justify the development of economically infeasible, environmentally troublesome and politically motivated water projects, including the Pa Mong Dam between Thailand and Laos (Jenkins 1968), the Pak Mun Dam in Thailand (WCD 2000), the Don Sahong in Laos (NCC 2013), the Lower Sesan II in Cambodia (Null et al. 2020), Yali Falls dam in Vietnam (Wyatt and Baird 2007), etc. Perhaps, the most well-documented case where the hydrological science was manipulated to securitize certain hydropower project can be found

in terms of the Xayaburi dam (Grünwald, Wang and Feng 2020). When the first official discussion about the Xayaburi dam in September 2010 started, the Laos government underwent a lengthy consultation process with the Mekong River Commission (MRC) and its members about the technical design to address potential negative impacts on downstream countries (Fox & Sneddon, 2019; Hensengerth, 2015). To ensure the feasibility of this project, the Laos government since 2007 hired several assistant companies (Pöyry Company, Andritz, Compagnie Nationale du Rhône) to monitor, evaluate and address various

concerns regarding the Xayaburi dam.

However, in October 2010, the International Centre for Environmental Management (ICEM) under the auspices of the MRC released a comprehensive hydrological study analysing the potential impact of mainstream dams, including the Xayaburi dam (ICEM 2010). The study concluded that many proposed Laotian dams on the mainstream may pose a significant impact on the water flow, and further studies need to be done to understand the scope and deepness of the negative implications on the

sustainability of the Mekong River (MRC 2011). The opposite attitude towards the planned Xayaburi dam came in August 2011 when the Laos government introduced the Pöyry Report stating that "Laos provided enough time and opportunities for evaluating the Xayaburi project" (Yasuda 2015: 117) and concluded that "Xayaburi dam follow the MRC design guidelines" for which there are no legal barriers to continue with the construction of the dam (PPLC 2011: 14). Despite the public outrage (Vandenbrink & Avary, 2012), unsuccessful lawsuits against assistant companies that did not act in the "good faith" (SFMG

2012, Finnish NCP 2013) and interstate interventions criticizing the unilateral development, particularly from Vietnam, Cambodia and the USA (Grünwald, Wang and Feng 2020), the Laos government formally continued with the construction of the dam in November 2012 anyway (Yasuda 2015). On the other hand, to calm down the public outcry and partially address the concerns from neighbouring countries, the Laos government decided to re-design part of its dam and invest over 100



million USD to ensure infrastructure safety against earthquakes and install new-type of turbines to mitigate negative impacts
for sediment flux and fish migration (Giovanni 2018). However, despite the commendable effort from the Xayaburi construction company, the MRC stated that there are insufficient information for evaluating the impact of the revised design changes and the unclear impact of modified fish passages (MRC 2019a). The controversies over the dam arose again when Laos started the trial operation test in July 2019. Thanks to the unfortunate coexistence of filling the Xayaburi reservoir with the scheduled maintenance of the Jinghong dam in China and ongoing severe droughts within the basin (Grünwald, Wang and

Feng 2020), the Xayaburi dam stirred a new wave of uncertainty over the future of the Mekong River.

While some observers consider the Xayaburi dam as a failure of the MRC mandate (Cronin and Weatherby 2015, Rieu-Clarke 2014), it was actually the Pöyry Report and strong lobby from multi-stakeholders (e.g. Pöyry, Andritz, Compagnie Nationale du Rhône, Thai Banks and Electricity Generating Authority of Thailand) that served as an ultimate tool for justifying the construction of the dam at all cost. Additionally, thanks to the strong determination of the Laos government in collaboration

with other multi-stakeholders to counter the existing knowledge about potential impacts of mainstream dams (see ICEM 2010), the Pöyry's Report clearly demonstrated how shallow socio-environmental impact assessment, selective evaluation and questionable technical-technological solutions (Pöyry 2011, WWF 2011) may outweigh the dominant water paradigms and completely change the perception of legal water governance instruments. In fact, after the breaking of the MRC Procedures for Notification, Prior Consultation and Agreement (PNPCA) which was not designed as a brake lever (i.e. prove or disapprove

the project) but rather as a tool to ensure transparency of ongoing mainstream projects (Hatda 2020), the MRC lost a significant amount of the funds and was forced by its donors to reduce the number of workers as a "punishment" for the final outcome with the Xayaburi dam (MRC, 2016). Last but not least paradox in the legitimization of the Xayaburi dam came to the attitude of several downstream countries. While Thailand's government strongly supported the dam construction to satisfy its future energy demand, it was also Thailand which time vehemently criticized the potential negative impacts on Thai society

(Grünwald, Wang and Feng 2020). Similar double-standard rhetoric can be also traced to Vietnam where Vietnam state authorities from strong opposition against Laotian hydropower dams to slowly move to conservative acceptance of changing the status-quo in order to not lose one of its strongest allies in Southeast Asia (Grünwald, Wang and Feng 2020). Such pragmatism was mainly driven by potential concerns about losing influence in Laos and a sort of self-preservation instinct to negotiate about the future Laotian projects.

The second type of the politicization of science is related to the spreading fake news (Tian and Liu 2016), creating misinformations (Biba, 2012, Biba 2016) and making other harmful interpretation of hydrological studies beyond the data (Embassy of the People's Republic of China in the Kingdom of Thailand, 2016; Embassy of the People's Republic of China in the Kingdom of Thailand, 2019). Among all politicized hydrological studies within the Lancang-Mekong Basin, there is the Eyes on Earth Study (EoE Study) published in April 2020 where authors claimed they found ultimate evidence about

manipulating the water flow by Chinese upstream dams (Basist & Williams, 2020). Such research conclusions attracted significant media attention and were intended to enforce better hydrological data transparency about mainstream dams. Within days, the New York Times as the first public media raised concerns about the potential impact of Chinese dams (Beech, 2020).



The medialization of the EoE Study findings were then followed by the Stimson Centre (Eyler & Weatherby, 2020) and other research commentaries from the MRC (MRC, 2020) and various researchers from Australia (Ketelsen, Sawdon and Räsären, 2020), Finland (Kallio and Fallon, 2020) and China (Tian et al. 2020) to official response from China's Foreign Ministry (Embassy of the People's Republic of China in the United States of America 2020) and the US Department of State (Pompeo, 2020). The accountable research discussion become under scrutiny since September 2020 when the political rhetoric both at official and public media emotionally escalated (Stilwell, 2020a, Stillwell, 2020b, PRC MOFA, 2020b) which subsequently caused the acute deadlock in the reasonable political dialogue between states.

However, due to (a) character of the research study (i.e. published by independent think-tank rather than the MRC and other well-grounded scientific institutions), (b) source of funding (i.e. funded by the United States Department of State in collaboration with the Mekong-U.S. Partnership (MUP) and other US stakeholders), (c) under-tested hydrological method (i.e. using the wetness index calculated upon the Sensor Microwave Imager/Sounder (SSMI/S) regression model rather than prevalent methods such as the Standardized Precipitation Evapotranspiration Index (SPEI) or the Standardized Precipitation Index (SPI)), (d) lack of rigorous research work (i.e. limited number of references and questionable peer-review process), (e) interpreting the firm assumptions beyond data (i.e. accusing Chinese dams as a main driver for hydrological changes without considering the cumulative effects of hydropower dams on the tributaries), and (f) refusing to address existing shortcomings of the study (i.e. lack of interdisciplinary discussion within official and semi-official research channels), the EoE study raised a lot of concerns among misinterpretation and politicization of science (Ketelsen, Sawdon and Räsären, 2020, Kallio and Fallon, 2020, MRC, 2020). Surprisingly, despite the theoretical-methodological flaws, the EoE Study soon become part of the public domain of the US Foreign Policy agenda and served as a political tool for increasing pressure on China's government (see Pompeo 2020, Stilwell, 2020a, Stilwell, 2020b). In contrary to the US political approach, China's government response to the EoE Study was relatively mild regarding (i) the general distrust to unknown research studies outside the traditional research channels (i.e. no letter of concerns or notification from the MRC), (ii) existing guarantees for sharing year-round hydrological data (i.e. re-emphasized during the 5th Lancang-Mekong Cooperation Foreign Minister Meeting in February 2020 and 3rd Lancang-Mekong Cooperation Leader's Meeting in August 2020), and (iii) outset of the Coronavirus Pandemic (COVID-19) (i.e. national priority was to tackle the virus, minimize the economic losses and keep funding the existing overseas projects) but after the intervention of the US government, China's state officials repeatedly assured to fulfil its commitments and ensure the sustainability of the river by deeper collaboration with the MRC (Embassy of the People's Republic of China in the United States of America, 2020, SCPRC, 2020).

While some observers perceive the EoE Study as a pretext for further involvement of the USA in Southeast Asia and accelerating the rivalry with China through the newly established Mekong-U.S. Partnership (MUP) and Quadrilateral Security Dialogue (Quad) (Xinhua 2020; U.S. Mission to ASEAN, 2020), the EoE Study undoubtedly (a) raised public awareness, (b) motivate China to keep its promises, and (c) encourage the USA to put more effort in transboundary water cooperation. Ironically, thanks to the EoE Study findings, the MUP in collaboration with the Stimson Centre launched the Mekong Dam Monitor to provide an interactive and publicly-available platform for investigating the hydrological changes (Basist et al.



2020). China's government welcomed this US initiative for putting more faith in scientific facts over words, preventing further politicization of water resources, and contributing to accountable research dialogue (Embassy of the People's Republic of China in the Kingdom of Cambodia, 2020; PRC MOFA, 2020a, 2020b). However, regardless of China's determination to advance water cooperation, there is still a lack of timely notification and reliable communication among riparian states. Such a trend can be specially identified after the scheduled maintenance of the Jinghong hydropower dam in July 2019 (MRC 2019b) and January 2021 (MRC 2021) which re-emphasized the general distrust towards Chinese commitments and spread public outcry in downstream countries.

## 3. Discussion and conclusions

The mismatch between scientific and non-scientific approaches towards hydrological studies is substantial in recent years. While the majority of water experts tend to be conservative and cautious in interpreting any water-related research findings, civil society, public media and politicians appear to be more emotional and creative in developing the arguments from the researchers. In fact, unlike scientists whose conclusions may be driven by self-interest (i.e. supporting narrative of funded governmental and other institutional research may pave the path to the job promotion) or matter of survivability (i.e. facing political persecutions and other obstacles in working life by sharing undesirable conclusions), individuals, independent journalists and politicians without scientific affiliation usually do not possess same code of conduct and responsibility for errors or other harmful impacts of their actions. To date, the politicization of science is mainly associated with climatology (i.e. discussion about the existence of climate change) and epidemiology (i.e. conspiration theories and hoaxes about COVID-19) but can be also traced to other research fields, including hydrology, environmental studies and social sciences.

By illustrating the existing dilemmas in socio-hydrology and complexity of anthropological factors affecting the perception of current water challenges, we claim that politicization of science (i) gives more power to non-scientists, (ii) undermines the trust in science and other research institutions, (iii) creates inequality among hydrological studies and water scientists, and (iv) provides more incentives for making research tailored to desirable outcomes. Our conclusions are based on the following trends. Firstly, more hydrological data nor the bigger inclusion of non-scientists into the research process does not necessarily produce a better understanding of the water uncertainties. Whenever scientists strife for getting more knowledge through data mining there is always an issue of wisdom that requires careful observation and time for analysing the recorded data. Although this process can be very difficult and frustrating considering the selection of relevant data and summarizing key observations in a timely manner, many researchers may during their investigations bounce on series of unexpected challenges and other limits enlarging the number of uncertainties that need to be addressed. Secondly, interpreting the hydrological studies through official and semi-official channels do not directly lead to the objective interpretation of hydrological data. As much as scientists from different research fields as well as state authorities responsible for transboundary water management may have a different view on the utilization of shared water resources, the attitude from civil society and the general public also significantly vary. For the majority of people, the most convenient and publicly visible source of information about ongoing water challenges



providing the mass media. Unlike the other sources, these unofficial channels use simplified facts and reiterate general
conclusions official and semi-official sources to grasp and develop the general understanding of complex water issues.
However, some of these information may possess a lot of errors or be (un)intentionally interpreted in certain ways to support
particular narratives which is why these sources should not be marginalized. Thirdly, if there is political will, there is no
difference between the quality of hydrological studies to justify desirable narratives. While there are many mechanisms on
how to deal with various errors, fake news, and other harmful misinformation, the misinterpretation of hydrological data may
arise both intentionally and unintentionally. Also, despite increasing transparency and multi-level verification of data, the
selective references on authors supporting certain narratives, gradual corruption of scientific institutions, gaps in accountable
interdisciplinary research dialogues and putting water issues into broader non-water context not only polarize the society but
also precipitate a deeper crisis of science.

This takes us to the politicization of science in the Lancang-Mekong Basin. By observing two case studies (Pöyry Report and
EoE Study), we found several common features. Firstly, both studies were used to serve to legitimize certain political actions
at all cost. Regardless of numerous theoretical, methodological and other research flaws, most of the references on such limits
were marginalized and interpreted as a "minor" problem. Whenever such overconfidence and high expectation for reconciling
the potential conflict of interests was fed by funders or personal convictions of investigators, it seems that actual conflict
resolution and hope for establishing new water cooperation was never an issue. In fact, while authors from both studies were
"open for discussion" to address existing concerns related to redesigning the Xayaburi dam and addressing existing theoretical-
methodological shortcomings of the wetness index, such proclamations remained unanswered and repeatedly denied which
keep escalating the conflict over facts among scientists. Secondly, both studies were used to counter previous hydrological
research and "plant seeds of doubt" about existing hydrological knowledge without considering the negative consequences for
both science and politics. Regarding the sociological sensitivity to any deteriorations of water in downstream countries, it
requires more effort to build mutual trust through cooperation than lose the faith in science by disparaging other scientists.
Because "science gathers knowledge faster than society gathers wisdom" (Asimov 1988), moving from official and semi-
official channels (i.e. multi-level water diplomacy) to the realm of mass media must be undertaken carefully (i.e. avoid to use
media as a sort of cementing own opinion) and professionally (i.e. consult with colleagues and get the consent from the
affiliated institution before publishing any research-based content) to minimize the emotional public outcry. Therefore, making
more official statements, campaigns and other denouncing activities to fight against misinformation may be futile without
bottom-up collaboration with grassroots societies that is crucial to restoring faith in science.

On the other, we claim that both studies possess several elementary differences. Firstly, there is an issue of incentives and the
purpose of the studies. While Pöyry Report was mainly designed to secure the construction of Xayaburi dam and intentionally
obstruct any legislative attempt for dam suspension, the EoE Study seems to unintentionally interpret its research conclusions
beyond data during the "quest" of finding the factors for altering the natural water flow (Basist and Williams, 2020). Secondly,
there is a different composition of multi-stakeholders supporting hydrological studies. Whereas Pöyry Report was mainly
supported by Laos and Thailand's state's authorities as well as hired assistant companies, financial institutions and other actors



involved in securing the Xayaburi dam, the EoE Study was mainly promoted by US officials, the Stimson Centre, civil society in downstream countries and pro-western media. Ironically, it was also the Thailand government which through the EGAT

nominally backed up the development of the Xayaburi dam and at the same time redirected the anger of the Thai civil society to the Laos government for continuing with the construction of the dam. A similar trait can be also found in terms of the EoE Study where most of the references in public media were put on account of the Stimson Centre Commentary (Eyler & Weatherby, 2020) rather than the original findings of the EoE Study. Thirdly, there is a difference in criticism of both studies. While Pöyry Report was the subject of numerous trials (Grünwald, Wang and Feng 2020) for causing harm and not acting in

the good faith, there was surprisingly no filled lawsuit on account of the authenticity of the EoE Study findings. On the other hand, while the interpretation of the Pöyry Report is quite straightforward (i.e. build or not to build the dam), there are more uncertainties coming up from the EoE Study findings on which there is no research consensus (i.e. cumulative impacts and climate change).

Our last goal was to discuss the unintended and unanticipated consequences of the politicization of hydrological studies.

Firstly, publishing unconventional, controversial and non-reviewed research results might get a new standard for future hydrological studies. Resigning on the accuracy of research findings, moral research code of scientists and sensational headlines with misinformation in public media should not be tolerated and may soon become dangerous precedence for undermining trust in hydrological science in general. In addition, if the research conclusions will remain to be predominantly interpreted by non-scientists rather than scientists with research history, such trend can easily become a "double-edged

weapon" and self-fulfilling prophecy for neo-Malthusian and other alarmists approaches invoking the looming water crisis (e.g. Bulloch a Darwish 1993). Secondly, both the Pöyry Report and EoE Study positively increase public awareness and motivated riparian states to establish deeper collaboration in transboundary water governance. While breaking the MRC PNPCA by Pöyry Report was a valuable lesson for the MRC that led to reconsidering the purpose of this mechanism (i.e. from build or not to build to ensure the transparency and accountable dialogue about proposed mainstream projects) and strong

proof for downstream countries governments that any coercive (i.e. using the US government as a "moderator with muscles" and threatening Laos government by damaging the relations with Vietnam) or legal actions (i.e. filled lawsuits) may not be effective as keeping the dialogue about other planned Laotian dams (e.g. Don Sahong, Luang Prabang, Sanakham), the EoE Study accelerated US efforts to find out the causes of Mekong River hydrological changes (i.e. establishment of the Mekong Dam Monitor) and motivated the US government to promote non-traditional security cooperation through newly established

MUP, including the good transboundary water governance. To conclude, socio-hydrology like any other interdisciplinary scientific fields face "identity insecurity" where possession of knowledge does not automatically implicate wisdom. As we mentioned above, the politicization of science is a complicated process of re-considering existing water paradigms and vigorous capacity to test alternative interpretation of complex phenomena. With growing medialization and misinterpretation of science, it is worthy to diversify research channels and put more pressure on reasonable scientific etiquette. Despite data

mining as well data processing might not always be perfect, sharing the hydrological data and discussing the potential implications among scientists and other multi-stakeholders is one of the well-established mechanisms to demyth the "toxic"



water narratives that distort the truth. While the quality of hydrological science may vary, we strongly believe that socio-hydrology may address these discrepancies and help to set the threshold for limits of scientific conclusions.

**Disclosure statement**

No potential conflict of interest was reported by the authors.

**Acknowledgement**

This research was funded by the National Key R&D Program of China (No.2016YFA0601604), Natural Science Foundation of China (No. 41701626 & No. 41661144044), and Post-Doctoral Research Project in Yunnan Province (YNBH19006), Yunnan Provincial Government project (W8163007).

**Author contribution**

WW and RG conceived, designed, and supervised the research. All authors contributed substantially to the writing and revising of the manuscript.

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
