# Peer review of "Misinterpretation of hydrological studies in the Lancang-Mekong Basin: drivers, solutions and implications for research dialogue"

_Hydrology and Earth System Sciences, 2021_

## Author Comment (AC1)

RC1

The study by Wang et al. with the title of "*Misinterpretation of hydrological studies in the Lancang-Mekong Basin: drivers, solutions and implications for research dialogue*" describes the politicization of hydrological science. The study then explains the drivers and intended consequences. Although the study makes good points in politizing the hydrological science, the paper will probably make the readers confused about how this research contributes to the special issue of "socio-hydrology and transboundary rivers." To make this contribution, the authors need to improve the literature review, which is currently very weak. Then, the submitted paper should mention the research gap. Thus, I highly suggest the submitted paper goes through a major revision at this stage with my following comment.

Thank you for your feedback. Based on the recommendations from you and the other reviewer, we (i) substantially revised the text, (ii) added the literature review, and (iii) added few tables as requested. For more details, see the comments below.

- My general comment is that the authors improve the writing of the submitted paper. In particular, please avoid using long sentences.

  **Fully accepted** – the text has been substantially revised throughout the manuscript and long sentences have been shortened.

- The abstract is highly weak in terms of how the submitted paper contributes to socio-hydrology and transboundary rivers.

  **Fully accepted** – the abstract has been revised and extended.

- I suggest removing the first sentence in the abstract. It has a loose connection with the rest of the abstract. Improve the last sentence in the abstract so that you can say

  **Fully accepted** – the first sentence has been deleted.

- The introduction needs to discuss the purpose of socio-hydrology and how this paper is connected to this purpose.

  **Fully accepted** – the introduction has been substantially revised to more reflect the connection between socio-hydrology and the politicization of science.

- Line 27: what do you mean by socio-hydrology perspective? How does this perspective help you with your research?

  **Fully accepted** – the text has been revised (see Abstract, Introduction and Chapter 1).

  Socio-hydrology considers science as a social process in which the water challenges cannot be solved in a purely technical manner and a plurality of both scientists and non-scientists is required to design more sophisticated water assessments.

- Line 39: typo error. The introduction is 1, and this title should be 2.

  **Fully accepted** – the text has been revised and changed.

- Lines 42-46: the authors already claimed that they use the socio-hydrology perspective. Thus, they should mention the difference between socio-hydrology and hydro-sociology. Accordingly, the reason they use socio-hydrology perspective.

  **Fully accepted** – the text has been revised (see Abstract, Introduction and Chapter 1).

- Line 48: improper references. Generally, the document really lacks a literature review on socio-hydrology. You may improve your literature review by the following studies and citing them:

  - Di Baldassarre, G., Viglione, A., Carr, G., Kuil, L., Salinas, J. L., & Blöschl, G. (2013). Socio-hydrology: Conceptualising human-flood interactions. Hydrology and Earth System Sciences, 17(8), 3295–3303. https://doi.org/10.5194/hess-17-3295-2013

  - Ghoreishi, M., Razavi, S., & Elshorbagy, A. (2021). Understanding Human Adaptation to Drought: Agent-Based Agricultural Water Demand Modeling in the Bow River Basin, Canada. *Hydrological Sciences Journal.*

  *- Elshafei, Y., Sivapalan, M., Tonts, M., & Hipsey, M. R. (2014). A prototype framework for models of socio-hydrology: Identification of key feedback loops and parameterisation approach. Hydrology and Earth System Sciences, 18(6), 2141–2166. https://doi.org/10.5194/hess-18-2141-2014*

  *- Gonzales, P., & Ajami, N. (2017a). Social and Structural Patterns of Drought-Related Water Conservation and Rebound. Water Resources Research, 619–634. https://doi.org/10.1002/2017WR021852.*

  **Fully accepted** – the references have been added (see References).

  Thank you for those very informative literature sources.

- Lines 48-49: "*many authors*" needs proper references.

  **Fully accepted** – the references have been added and the text has been revised.

- Lines 49-50: what do you mean by calling uncertainty, politics, and power as variables?

  **Fully accepted** – the text has been revised and integrated into Chapter 2 and Chapter 5 (Current challenges and dilemmas in depoliticization of science; Discussion and conclusions).

  We investigated the standard research procedures to better understand the reliability of science. We focused on the biases in the peer-review process, the dichotomy between quantity and quality of the research studies, social and power inequality between multi-stakeholders, limits of the fact-checking tools and other verification mechanisms.

- Line 51: it seems that the reference is irrelevant to this sentence. You may cite the following study:

  - Wei, J., Ghoreishi, M., Souza, F., Lu, Y., & Tian, F. (2020, May). Socio-hydrological approach to understand conflict and cooperation dynamics in transboundary rivers. In *EGU General Assembly Conference Abstracts*(p. 7148).

  **Fully accepted** – the text has been revised and more references have been added.

- Lines 51-52: proper references for "To date … assumption"

  **Fully accepted** – the text has been revised and proper references have been added.

  To date, there can be identified many water paradigms which are shaped by self-fulfilling theories and other socially constructed assumptions (see Zeitoun et al. 2017; Earle, Jägerskog and Öjendal, 2010; Zeitoun and Mirumachi, 2008; Yoffe, Wolf, and Giordano, 2003; Gleick, 1998).

- Lines 52-59: an example of a long sentence that confuces the readers.

  **Fully accepted** – the text has been shortened and revised.

- Lines 60-62: ver unclear sentence.

  **Fully accepted** – the sentence has been deleted.

- Line 117: if you mean baker (2021) by "authors", use "the author"

  **Fully accepted** – the text has been changed.

  According to Baker's research work exploring the politicization of science, at least half of the scientific papers can be trusted (Baker, 2016).

- Lines 124-126: not clear why "full trust is troublesome"

  **Fully accepted** – the text has been revised and added.

  We divided the reasons into three sub-chapters (2.1. Actors and peer-review process; 2.2. Uncertainty and limits of science; 2.3. Conclusions and research mindset) to provide a better explanation of the current dilemmas in the politicization of science.

- Line 334: At the end, the conclusion leaves the readers what the contribution of this paper to socio-hydrology is. I highly suggest adding a section to clarify this point.

  **Fully accepted** – the text has been revised and added.

  We divided the "Discussion and Conclusions" into three sub-chapters to reflect each of the research question and application of the scientific findings on the two case studies (Pöyry Report and Eyes on Earth Study).

---

## Author Comment (AC2)

RC2

This paper focuses on the politicization of hydrology science in transboundary Lancang-Mekong Basin. It analyses the drivers of politicization, solutions for addressing the misinterpretation, and takes the Lancang-Mekong as an example to explain the politicization. The politicization in transboundary river basin plays an important role in the evolution of cooperation and conflicts, and this paper gives us new insights into this important issue. However, the paper seems more like an opinion paper instead of a research article. Although the manuscript reviews the extant studies of socio-hydrology, the relationship between misinterpretation of hydrological science and socio-hydrology is still elusive and far-fetched. This paper needs major revisions as listed below in detail.

*Thank you for your feedback. Based on the recommendations from you and the other reviewer, we (i) substantially revised the text, (ii) added the literature review, and (iii) added few tables as requested. For more details, see the comments below.*

1. The introduction part is too simplified to notify us why it is important to conduct this research, what the other researchers have done in this field, and what is the difference between this study and the extant ones. Now we cannot find out the significance of this study, and whether the theory in this manuscript is first proposed or purely an extension of extant theories.

   **Fully accepted** *– the abstract has been revised and extended.*

2. In the introduction part, it is stated that "*what feeds the uncertainty the most is the interpretation beyond data, different quality and purpose of the hydrological studies that may create a series of controversies and polarize both scientific and non-scientific audience.*" This statement without any reference or explanation seems very causal. More evidences are needed to reach this conclusion.

   **Fully accepted** *– the sentence has been deleted.*

   *We revised the text and put more explanation with references in Chapter 2 (Drivers and dilemmas in depoliticization of science).*

3. The section of "*socio-hydrology and politicization of science*" seems not to clarify why socio-hydrology is relevant to politicization of science. Since this special issue is about socio-hydrology, the clarification is very important. This section now seems to introduce the concept of socio-hydrology and the representation of politicization of science separately, instead of including politicization of science into the scope of socio-hydrology, which makes the narrative far-fetched.

   As stated in the manuscript, socio-hydrology focuses on the human-water interactions and the resulting "emergent behavior", particularly the mechanism that human not only affects hydrological system, but also responds to its variability. However, the only "interaction" mentioned in this section is the interaction between scientists and non-scientists. The section mainly introduces the causes of misinterpretation (scientists v.s. scientists, scientists v.s. non-scientists), and several groups of non-scientists. This narrative informs us that non-scientists could take effect in transboundary river management, including volunteers, politicians, and subversive actors, yet does not explain how they can affect the interactions between human and water, or how politicization of science affects the human-water interactions.

   **Fully accepted** *– the text has been fully revised and added*

We re-wrote Chapter 1 (Socio-hydrology and politicization of science) to outline the conflict of interests between policy-makers, scientists and civil society and examine the connection between the politicization of science and socio-hydrology.

We also added more explanation how civil society is useful to replicate the local knowledge and co-design better adaptation strategies for water resources management. The conflict of interests between interdisciplinary scientists was put in Chapter 3 (Solutions for accountable research dialogue).

4. The section of "*drivers, techniques and degrees misinterpretation of hydrology science*" gives us three drivers of misinterpretation of hydrological science. As mentioned before, are these three drivers first proposed in this manuscript or transformed from previous studies? How to verify these three drivers are complete or main drivers of misinterpretation?

**Partially accepted** – the text has been fully revised and added

The drivers were derived from the literature review related to the theories of politicization of science, socio-hydrology, water resources management, and transboundary water governance (see References). To better understand the drivers of the misinterpretation of hydrological findings and conceptualization of the politicization of science, we fully re-wrote the "Conclusion and Discussion" chapter and demonstrated the current water challenges on two case studies (Pöyry Report and Eyes on Earth Study).

The compilation of the drivers and application of these concepts in the context of socio-hydrology was made for the first time by authors. To date, there is no wide consensus about what drivers contributes to the misinterpretation of hydrological science. Also, compared to other misinterpreted research fields (see Pardini et al. 2021; Sarewitz, 2015), there is limited literature exploring the politicization of hydrological science in detail.

For this purpose, we are determined to conduct further studies to explore the information gaps and design the Uncertainty Interaction Checklist (see Subchapter 5.3. Politicization of science and future pathways for the socio-hydrology) to address the misinterpretation of science.

The section of "*solutions for de-politicization of science*" introduces three channels, including official channels, semi-formal communication channels, and informal communication channels. However, the manuscript only introduces the features and forms of different channels which already exist, instead of putting up with practical solutions to solve the problem of misinterpretation. Therefore, it is inappropriate to define this section as "solutions".

**Fully accepted** – the headline has been rewritten and text has been added

The information from the Chapter 3 (Solutions for accountable research dialogue) has been applied in two subchapters – 5.2. Political implication of the misinterpreted hydrological studies; 5.3. Politicization of science and future pathways for the socio-hydrology) in the "Conclusion and Discussion" section.

5.  The section of "*politicization of science in the Lancang-Mekong Basin*" takes two examples including Pöyry Report and EoE Study to explain the misinterpretation in the Lancang-Mekong. Two main drivers for the politicization of hydrological science are identified. The analysis of the two cases should be imbedded in the framework of the two former sections, so that the analysis framework mentioned before can be proved effective.

    **Fully accepted** – the text has been fully revised and added

    We divided Chapter 4 (Politicization of science in the Lancang-Mekong Basin) into two sub-chapters (4.1. Xayaburi dam and the Pöyry report; 4.2. Chinese mainstream dams and the Eyes on Earth Study) to better explain the misinterpretation and politicization of the hydrological studies in the Lancang-Mekong Basin.

    The analysis of both case studies has been then embedded in the "Conclusion and Discussion" for a better understanding of the connection between the politicization of science and socio-hydrology.

6.  In the part of discussion, the two cases of misinterpretation in the Lancang-Mekong are compared, and the intended consequences of the politicization of hydrological studies. However, it still lacks the discussion on how these factors affect human-water interactions.

    **Fully accepted** – the text has been fully revised and added

    Further discussion about both case studies has been then clarified in three sub-chapters (5.1. Actors and conflict of ideas over the transboundary water resources; 5.2. Political implication of the misinterpreted hydrological studies; 5.3. Politicization of science and future pathways for socio-hydrology).

7.  The main innovation should be clarified and compared to extant studies in the discussion part. Is the framework of analysis of drivers/solutions of misinterpretation and its application in the Lancang-Mekong the main innovation?

    **Fully accepted** – the text has been fully revised and added

    We highlighted the innovations in sub-chapter 5.3. Politicization of science and future pathways for socio-hydrology.

    We draw the innovations on the case studies and highlighted several solutions for establishing the accountable research dialogue. Firstly, we recommend to conduct better inter-institutional collaboration by setting up minimum guidelines for reliable hydrological science within the MRC PNPCA and consulting all-new water evidence with the MRC. Secondly, we encourage interdisciplinary scientists to discuss better fact-checking tools and the idea of the Uncertainty Interaction Checklist that will require further consideration. Thirdly, we highlighted that the politicization of science creates both positive and negative implications for transboundary water governance. However, caution with the interpretation of the hydrological findings and better engagement of the civil society in the research process is advised.

8.  The title numbers of sections now is disordered.

    **Fully accepted** – the number of sections has been fully revised

9. The language should be modified by an English native speaker, since there are many grammatical errors and typos. The word "quantitate" in line 47 is a typo. The "presents" in line 62 should be "present". The title in line 115 is not right. The "strife for" in line 350 should be "strive for". The sentence in line 359 is wrong. The "on the other" misses a word "hand" in line 387.

**Fully accepted** – the text has been fully revised, including the Reference section

10. There are many sequence numbers in the manuscript, and some of them are listed as (a)(b)(c), while the others are listed as (i)(ii)(iii), which should be unified and reduced.

**Fully accepted** – the listed order has been reduced in the whole text

11. Annotations in brackets are abundant, which aim to explain the formal concepts or sentences. But too many annotations increase difficulty to read.

**Fully accepted** – the annotations in the brackets have been reduced

12. Figures or tables could be used to visualize the framework of drivers/solutions/consequences of misinterpretations of hydrological science.

**Partially accepted** – the tables have been added

We added three tables related to (i) opportunities and challenges for accountable dialogue, (ii) drivers of the politicization of science, and (iii) a brief comparison of the politicization of the Pöyry Report and Eyes on Earth Study.

We left the solutions in sub-chapter 5.3. (Politicization of science and future pathways for the socio-hydrology) in the text without summarizing in another table because we want to furtherly conceptualize the Uncertainty Interaction Checklist (UIC) in the following research paper.